# Probiotics and Postbiotics as the Functional Food Components Affecting the Immune Response

**DOI:** 10.3390/microorganisms11010104

**Published:** 2022-12-31

**Authors:** Aleksandra Szydłowska, Barbara Sionek

**Affiliations:** Department of Food Gastronomy and Food Hygiene, Institute of Human Nutrition Sciences, Warsaw University of Life Sciences (WULS), Nowoursynowska St. 159C, 02-776 Warszawa, Poland

**Keywords:** probiotics, postbiotics, functional food, immune response

## Abstract

The food market is one of the most innovative segments of the world economy. Recently, among consumers there is a forming trend of a healthier lifestyle and interest in functional foods. Products with positive health properties are a good source of nutrients for consumers’ nutritional needs and reduce the risk of metabolic diseases such as diabetes, atherosclerosis, or obesity. They also seem to boost the immune system. One of the types of functional food is “probiotic products”, which contain viable microorganisms with beneficial health properties. However, due to some technical difficulties in their development and marketing, a new alternative has started to be sought. Many scientific studies also point to the possibility of positive effects on human health, the so-called “postbiotics”, the characteristic metabolites of the microbiome. Both immunobiotics and post-immunobiotics are the food components that affect the immune response in two ways: as inhibition (suppressing allergies and inflammation) or as an enhancement (providing host defenses against infection). This work’s aim was to conduct a literature review of the possibilities of using probiotics and postbiotics as the functional food components affecting the immune response, with an emphasis on the most recently published works.

## 1. Introduction

Food is a fundamental human requirement and meets the nutritional needs of an individual. The use of functional food is a method of preventing diseases and improving the health of civilization and is becoming more popular and widely consumed, and in highly developed countries, the functional food sector is growing much more rapidly than other segments of the food market. The term “functional food” first appeared in Japan in the 1980s [1]. There are many definitions relating to the term “functional food”. They include foods placed on the market with health claims. However, over the years, there were a lot of definitions ascribed to functional foods [2,3,4,5,6]. The most current version of the definition, according to the Functional Food Center (FFC), defines “functional foods” as “Natural or processed foods that contain biologically-active compounds; which, in defined, effective, non-toxic amounts, provide a clinically proven and documented health benefit utilizing specific biomarkers, to promote optimal health and reduce the risk of chronic/viral diseases and manage their symptoms”. This definition is distinctive because it recognizes “bioactive substances”, or biochemical molecules that enhance health via physiological mechanisms. The mission of the FFC is to harmonize the definition of functional food to legitimize functional food science, improve the marketing of functional products, and improve international communication in this respect [7].

Functional foods cover a wide range of product types. Functional foods are categorized depending on the type of food or the active ingredients that are either added or naturally present in the food, including vitamins, fiber, flavonoids, minerals, fatty acids, carotenoids, and others. In addition to low-cholesterol, low-energy, high-protein foods, functional probiotic foods and innovative functional products with the addition of postbiotics can be distinguished [8]. In the literature, several reports can be found describing the fact that various food components such as glucans, polyphenols, probiotics, prebiotics etc., which can affect the immune response [9,10,11,12,13,14]. The human microbiome consists of a population of microorganisms such as bacteria that populate the digestive tract. The composition of the intestinal microbiome includes so-called “probiotic bacteria”. Probiotic microorganisms are defined as “live microorganisms that, when administered in adequate amounts, confer a health benefit on the host” [15].

To be able to talk about the beneficial effects of probiotic microorganisms on human health, they ought to be able to endure the harsh circumstances of the human stomach and digestive system. This means the capability of the probiotics to survive the passage through the upper gastrointestinal tract (GT), to withstand the gastric juices and bile salts, and to multiply, colonize, and function in the gut. Probiotics capable of modulating the immune response in the host’s body are called immunobiotics [16]. The probiotics have to remain functional after passing through the digestive system to carry out the desired action and provide the desired health benefits. A properly functioning immune system is the basis for maintaining homeostasis in the human body. The intestinal microbiota and probiotics are composed of mixed cultures of living microorganisms that can positively affect human health through their antibacterial, antiviral, anti-inflammatory, and immunomodulatory effect. Dysbiosis is the abnormal composition of the intestinal microbiome observed in a variety of pathological conditions Pathogenic microbial attack-induced host gut dysbiosis may lower the total probiotic bacterial load in the host gastrointestinal tract, which may lead to inflammation and secondary infections [17]. The gut microbiota is important not only for nutrient digestion and absorption, but also for the homeostatic, maintenance of the gut barrier, metabolism, and host immunity [18,19]. It has been reported that the dysbiosis play some role in the pathogenesis of many diseases such as gastrointestinal, cardiovascular diseases, obesity or diabetes [20,21,22,23].

However, advances in food science have contributed to the discovery of a new non-viable form of probiotics. They were frequently referred to by several names, such as postbiotics, paraprobiotics, heat-inactivated probiotics, or ghost-biotics, to describe inanimate microorganisms and/or their components that confer beneficial properties to one’s health [23,24]. Postbiotic is a term derived from the Latin word ‘post’, meaning after, and ‘bios’, from the Greek word meaning life. Further, the ‘biotic’ family of terms (probiotics, prebiotics, synbiotics, and postbiotics) concerns microbes (or their substrates) [21,22,24,25].

The International Scientific Association for Probiotics and Prebiotics (ISAPP) defined a postbiotic as “a preparation of inanimate microorganisms and/or their components that confers a health benefit on the host”. This definition is very well-developed and can provide the basis for the accurate exchange of information on postbiotics between the science world, the food industry, and the legislative body [20].

Postbiotics are soluble factors (products or metabolic by-products), secreted by live bacteria, or released after bacterial lysis, such as enzymes, peptides, teichoic acids, peptidoglycan-derived muropeptides, polysaccharides, cell surface proteins, and organic acids. It might also offer physiological benefits to the host by providing additional bioactivity [23,26].

The methods for manipulating the human microbiome include diet, microbial supplements such as probiotics or prebiotics, and antibiotic-based microbial suppression techniques. Current trends in metagenomic analysis or genome sequencing will help to improve knowledge about these commensal microbes and draw attention to the unique characteristics of the microbiome [27].

It is important to mention that a novel concept has recently emerged in science—the “personalized microbiome”, which is based on an individually composed, personalized diet [28,29]. Through various technological processes, including lactic acid fermentation, commonly consumed foods can be transformed into so-called “functional foods”, which demonstrate greater consumer acceptance than pharmaceutical supplements [27,30].

Therefore, the objective of this work was to provide a literature review of the possibilities of using probiotics and postbiotics as functional food components affecting the immune response, with the emphasis on the most recently published works.

## 2. Probiotics and Host’s Immune System

The immune system has a significant impact on the pathogenesis of several diseases. Its main function is to defend the body against various pathogens by recognizing “danger” (damage-associated molecular patterns (DAMPs)) and “stranger” (pathogen-associated molecular patterns (PAMPs)) molecular motifs according to the danger theory [31,32,33].

### 2.1. Concept of Probiotics

The concept of “probiotics” is closely related to “microbiota”. This term refers to the microorganisms such as bacteria, fungi, viruses, and protozoa, that exist in the human gastrointestinal tract. However, the disturbances in the composition and functions of these microorganisms are referred to as ‘dysbiosis’. The etiopathogenesis of many illnesses is connected to microbiota impairment. Currently, methods of microbiota modification, including the use of probiotics, are gaining in popularity due to the role they play in the etiopathogenesis of many illnesses. Probiotics are able to stimulate and modulate the immune response regardless of their viability [34].

### 2.2. Mucosal Immune System

The probiotic bacteria show stimulating activity of the immune system of mucosal membranes (mucosa-associated lymphoid tissue (MALT)), also the so-called common mucosal immune system (CMIS). The MALT system includes elements such as parts of the gastrointestinal immunity system (gut-associated lymphoid tissue (GALT)), the respiratory system (bronchus-associated lymphoid tissue (BALT)), the genitourinary system (genitourinary-associated lymphoid tissue (GUALT)), but also the skin (skin-associated lymphoid tissue (SALT)), although another expression for that set of cells is the skin immune system (SIS). The digestive tract is colonized by numerous commensal microorganisms, which make up the so-called microflora, particularly abundant in the intestine. These microorganisms, mainly bacteria, are estimated to be 10^14^ cells, interacting with the mucous membrane lining the digestive tract and constituting an important line of defense against pathogens from the external environment (bacteria, viruses, fungi, parasites). The human body has developed a complex lymphatic tissue system associated with bowel mucosa (GALT) to combat infectious and potentially harmful agents entering the gastrointestinal tract. This arrangement includes, in a comprehensive manner, both lymphatic structures directly linked to the intestinal mucosa, as well as the Peyer patches, lymph follicles, and mesenteric lymph nodes. The importance of GALT for the proper functioning of the host immune systems is emphasized by the fact that this system accounts for more than 75% of lymphatic cells of the entire immune system, including about 50% of lymphocytes, and the production of around 80% of all immunoglobulins, in particular IgA antibodies, which are separated into the mucous membranes and are called secretory IgA (SIgA). These antibodies are responsible for capturing antigens and preventing them from passing through the mucous membranes into the body. The presentation of the antigens to the immune system’s immune response in organized lymphatic follicles of the intestinal mucosa associated with the GALT system shapes the immune response by deciding whether to trigger an inflammatory reaction or tolerance to a specific antigen [35,36].

The intestinal immune system has developed two arms of adaptive anti-inflammatory defense that typically protect the epithelial barrier: (1) immune exclusion carried out by secretory IgA (SIgA) and IgM (SIgM) antibodies to control colonization of microorganisms and dampen penetration of potentially harmful antigens; and (2) suppressive mechanisms to prevent hypersensitivity to innocuous antigens, particularly food proteins and the commensal microbiota. The latter phenomenon (oral tolerance) is mostly reliant on regulatory T (Treg) cells induced in mucosa-draining lymph nodes where dendritic cells carry exogenous luminal antigens and become conditioned for stimulation of Treg cells. Pentameric IgM and locally synthesized polymeric IgA, primarily dimers, are exported by the epithelium to strengthen the mucosal surface barrier by the polymeric Ig receptor ((pIgR) or membrane secretory component). Inside the epithelial cells, secretory antibodies can also serve this purpose. The fact that probiotics affect the various components of the GALT immune system not only results in the secretion of these many active substances, e.g., cytokines, immunoglobulins, growth molecules, including such substances as lysozyme and defensin, but also affect and increase the renewal of cells in the gastrointestinal tract [37,38].

### 2.3. The Immune Regulation by Probiotics

The probiotic bacteria activate the anti-inflammatory and regulatory action of the immune system due to intestinal infection. It is essential to select an appropriate probiotic microorganism strain and to determine the appropriate dose and duration of dosing, which is crucial for achieving the desired effect and the immunological status of the host microorganism [39].

In the immunomodulation, probiotic antigenic fragments, such as cell wall compounds, have the ability to cross the intestinal epithelial cells and M cells in Peyer’s patches and then to modulate the innate and adaptive immune responses in the body [40].

However, the immunomodulatory properties of probiotics are also due to the release of cytokines such as interleukins (ILs), tumor necrosis factors (TNFs), transforming growth factor (TGF), interferons (IFNs), and chemokines from immune cells (epithelial cells, lymphocytes, granulocytes, macrophages, mast cells, and dendritic cells (DCs) which further regulate the immune system [41,42].

The mechanism of immune regulation by probiotics is shown in Figure 1.

Probiotics contribute to the proper functioning and differentiation of basic elements of the immune system cell populations such as dendritic cells, macrophages, T-lymphocytes, and B-lymphocytes. Probiotic microorganisms have a multidimensional effect on the processes of intestinal immune cells. The inflammatory process depends on proinflammatory and anti-inflammatory cytokines, where the anti-inflammatory cytokine, interleukin-10 (IL-10), is produced by monocytes, T cells, B cells, macrophages, natural killer cells, and dendritic cells, which inhibit the proinflammatory cytokines, chemokines, and chemokine receptors, that are responsible for intestinal inflammation [43].

The immune activity of probiotics depending on the interleukin produced. The immunoregulatory probiotics characterized the production of IL-10 and Treg cells, which leads to a reduction in IBD, autoimmune diseases, and allergies. On the other hand, in the case of immunostimulatory probiotics, IL-12 is produced, which can activate NK cells and develop Th1 cells. As a result, infections, cancers, and allergies are reduced (Figure 1).

### 2.4. Effect of Immunomodulation by Selected Probiotics

The previous works regarding the effect of immunomodulation by probiotics as food components are shown in Table 1.

One of the criteria taken into account for the evaluation of probiotic microorganisms is their safety of use. Probiotics should have the status of being generally recognized as safe (GRAS), which means that they should be generally accepted as safe [52].

Many studies have demonstrated the positive health effects of probiotics [53,54,55,56,57]. It was found that different strains of bacteria, although belonging to the same species, may have different effects on the organism. Therefore, a proven clinical benefit should be considered when selecting the bacteria used in food or medical supplements. Most of the currently available probiotic products are in the diet supplement status, not the medication status. The results of the inspections carried out indicate a number of discrepancies between the declared content and the actual content [58]. Many organizations, including the European Society for Pediatric Gastroenterology, Hepatology and Nutrition (ESPGHAN), address the issue of the supervision of probiotics production. Probiotic bacteria have been shown to affect the anatomical, physical, and microbiological barriers of the gastrointestinal tract as well as macro-organism resistance, including the resistance of the local digestive system, through colonization in the host intestine. This has been found, among other things, to reduce the risk of growth of potential pathogenic bacteria [59].

Probiotics, due to their positive effects on the immune system, have been found to be used in many pathological conditions, such as gastrointestinal infections [37,60], diarrhea including infectious and antibiotic-related diarrhea [61], allergic diseases [62,63], cancer [64,65], and infectious diseases of the respiratory and genitourinary system [66,67].

It is worth mentioning that, in view of the worldwide epidemiological situation with regard to the novel Coronavirus disease-2019 (COVID-19), information on the immune system has been given a great deal of attention. According to the WHO report, as of 16 January 2022 over 323 million confirmed cases and over 5.5 million deaths had been reported worldwide. Probiotics have been shown to restore stable intestinal microflora by regulating innate and adaptive intestinal immunity, making them useful in the fight against this disease [68,69].

However, due to the use of probiotics in certain specific circumstances, i.e., in patients with marked immune disorders. (e.g., congenital or acquired immune disorders, patients on immunosuppressive therapy), and in those with minor defects (e.g., intestinal barrier damage, usage of broad spectrum antibiotics), various adverse effects such as bloating, bacteremia, or fungemia have been described, and the possible transfer of resistance genes to antibiotics between bacteria [70,71].

## 3. Post-Immunobiotics

### 3.1. Immunomodulatory Effects of Postbiotics—General Remarks and Health-Promoting Benefits

Postbiotics may be included in functional food ingredients due to the variety of health properties they exhibit. Although the importance of postbiotic pro-health was initially overlooked and underestimated, over time, more and more scientific evidence has become available. The complex mechanisms of postbiotic effects on the body are currently being studied intensively [72,73,74,75]. Most of the studies carried out so far were carried out with different strains of *Lactobacillus* [73,74].

Postbiotics can be divided either according to the basic composition of the lipid (e.g., butyric acid, propionic acid), protein (e.g., lactalepine, p40 protein), carbohydrate (e.g., galactopolysaccharides, lipoteichoic acid (LTA), vitamin/coenzyme (e.g., B vitamins), the presence of organic acids (e.g., propionic acid), and more complex compounds such as polypeptides from peptides, or based on their physiological functions such as immuno-inflammatory action, anti-obesity-resistant, cholesterol-reducing, reducing hypertension, anti-cancer, or anti-oxidative function [76,77,78].

Non-viable microbial cells, their molecules, residues from cell walls or cell lysates, factors produced by probiotic microorganisms, and endogenous components formed in the microorganism–host interaction are active biological elements that, in the same way as probiotics, can bring potential health-promoting benefits for the host [24]. Cellular and molecular mechanisms are involved in postbiotics’ influence on immune system and potential pro-health profits. The stabilization of the host–microbiome balance through the preservation and strengthening of the integrity of the intestinal mucosa barrier is of key importance regarding the protection against infections and for the inhibition of the development of pathogens [79]. The benefits of postbiotics stem from their direct effect on the digestive system, which includes relief of symptoms of gastrointestinal disorders such as bloating, and infectious and antibiotic-related diarrhea. Therefore postbiotics can be useful in the therapy of gastrointestinal disorders, such as ulcerative colitis, irritable bowel syndrome, or necrotizing enterocolitis [80]. The pleiotropic activities of postbiotics include their influence on immunological resistance as well as anti-inflammatory, anti-oxidant, and anti-cancer properties [81]. Due to the immunomodulatory effect, the indirect impact on bidirectional communication between the intestines and diverse body systems and organs, postbiotics can impact on various diseases. Changes in the components of the intestinal microbiome affect the immune response and homeostasis in the respiratory tract and on lung health or disease [82,83]. Although the underlying mechanism of the gut–lung axis is still uncovered the most prominent immunomodulatory metabolites are considered to be short-chain fatty acids [84,85]. Similar interactions of the gut microbiome on the brain are reported in the literature. The mechanism of the gut–brain axis, despite advancements in understanding, is also unclear. The postbiotics molecules, mainly short-chain fatty acids, which are lipophilic molecules, can cross the blood–cerebrospinal fluid barrier and can reach the regions of the central nervous system responsible for immune regulation [86]. The second potential immunomodulatory mechanism by which the gut microbiota influence the central nervous system is the transmission of signals by cytokines [87,88].

Local postbiotic action in the gastrointestinal tract is accompanied by systemic interactions within the intestinal and other organs (Figure 2). Therefore, similar to probiotics, a potential positive postbiotic influence can be seen in a broad spectrum of illnesses including cardiovascular and respiratory system diseases, in the treatment of allergic and dermatological diseases, metabolic, neurological, and even psychiatric disorders [81]. Examples include beneficial effects on inflammatory bowel disease (IBD), and non-infective persistent diseases such as obesity, type 2 diabetes, cancer, atopic eczema, atopic dermatitis, the healing of burns and scars, and on neurological and developmental disorders (Parkinson’s disease, Alzheimer’s disease, autism, multiple sclerosis) [80].

The range of metabolites formed during the interaction of probiotic microorganisms with their host is extensive [79]. The pleiotropic activity of postbiotics, and the widespread distribution of substances, components, and cells indicate a wide potential for their applications.

### 3.2. The Postbiotics’ Impact on Immune Homeostasis

Human immunity is strongly related to the state of the intestinal microflora, which determines the maintenance of immune homeostasis. The innate and adaptive immune system is supported and controlled by signals transmitted by gut microbes providing the defense and protective responses against various pathogens [89]. During infection, pro-inflammatory (i.e., TNF-α, IL-1*β*, and IL-6) or anti-inflammatory (i.e., IL-10) cytokines are produced and released from different types of cells within the immune system. In the response of the immune system to the disease stimulus, it is crucial to maintain balance, as both excessive and inappropriate reactions can be unfavorable. Inflammation is mainly divided into acute and chronic. In an acute inflammatory reaction, infectious agents are removed first, followed by the repair stage [90]. Chronic inflammation occurs when, despite the lack of a external factor, inflammatory reactions still occur. Chronic inflammation leads to damage of the tissues and organs [91].

The immunomodulatory effects of postbiotics on innate and adaptive immunity result from the effects on toll-like receptors (TLRs) and on nucleotide-binding oligomerization domain-containing protein (NOD)-like receptors (NLRs) [73]. Toll-like receptors are transmembrane cellular proteins that enable the recognition of various pathogens. They are responsible for initiating the immune response. Eleven types of toll-like receptors have been distinguished [92]. TLR2 has been found to be responsible for the inflammatory response caused by Gram-positive bacteria and TLR4 for the reaction in the case of Gram-negative bacteria [80,93]. Ménard et al. (2004) [94] investigated the anti-inflammatory properties of postbiotics from two different lactic acid bacteria (*Bifidobacterium breve* and *Streptococcus thermophilus*) on colon cancer cells. They showed an increase in the proinflammatory cytokines level and a decrease in the release of TNFα, which was the result of interaction with a toll-like receptor that has the ability to recognize related pathogens. Individual TLR receptors can bind to a specific bacterial structure: lipopolysaccharides are identified by TLR4, lipoproteins, lipoteichoic acid, and peptidoglycan are identified by TLR2, and flagellin is identified by TLR5. The second mechanism of the immunomodulatory action of postbiotics takes place through the nucleotide receptors of the oligomerization domains, thanks to which it is possible to recognize different ligands of pathogenic microorganisms. As a result of NLRs’ activation, the pro-inflammatory cytokines interleukin (IL)-1β and IL-18 are secreted by the intestinal endothelial cells and cells of the immune system [95]. Multiprotein complexes called inflammasomes, in particular NLRP3 and NLRP1, are crucial for the maintenance of the integrity of the intestinal barrier and the adaptive and innate immunity of the intestinal epithelial cells and immune system cells [96].

### 3.3. Postbiotics—The Role of Gut–Organ Axis on the Immune System Modulation

Some researchers see additional benefits of postbiotics from their easier penetration through the mucosal barrier into cells, while living probiotic microorganisms need adhesion to intestinal epithelial cells to modulate the immune response. Individual components of postbiotics, such as: exopolysaccharides, teichoic acids, lipoteichoic acid, peptidoglycans, antioxidant enzymes, and short-chain fatty acids including acetic, propionic, and butyric acids can effectively modulate the immune response and, through toll-like receptors, exert anti-inflammatory and stimulating effects on the immune system [78,97]. Short-chain fatty acids are one of the most important gut microbe-derived components. By influencing gut–organ interactions (i.e., the gut–lung axis, gut–brain axis) they seem to be key mediators for setting the tone of the immune system (Figure 2) [81,98]. Short-chain fatty acids can also directly affect immune cells by modifying cell signaling and epigenetic regulation [98]. Nagai et al. (2006) [99] demonstrated that signaling through TLR2 and TLR4 promotes hematopoiesis stimulating the innate immune system. Short-chain fatty acids pass from the gut into systemic circulation and reach the bone marrow where they seem to influence myelopoiesis [82]. Butyrate has anti-inflammatory properties through the inhibition of proinflammatory cytokines, which activates nuclear factor-kappa involved in immune and inflammatory reactions. Another potential pro-health effect of butyrate is its anti-cancer effect, which is a result of the histone deacetylases blockade influencing the gene expression in colon cells.

Exopolysaccharides have the unique ability to prevent or control infection by forming protective biofilms on intestinal epithelial cells and reducing or inhibiting film formation by pathogenic bacteria [100,101]. Exopolysaccharides have various bioactivities such as immunomodulatory, anti-inflammatory, anti-tumor and anti-mutagenicity, antioxidant (enhancement of antioxidant enzymes activities such as catalase, glutathione peroxidase, and superoxide dismutase), anti-bacterial, and anti-viral effects. Additionally, they can inhibit cholesterol absorption. The essential structural components of the cell walls of Gram-positive bacteria are lipoteichoic and teichoic acids. They exhibit several properties, including anti-cancer, immunomodulatory, and antioxidant properties [101,102,103,104].

Lipoteichoic acid promotes the non-specific anti-inflammatory response by releasing anti-infectious peptides (defensin and cathelicidin) [81]. In a study by Zadeh et al. (2012) [105], they found that lipoteichoic acid caused an excessive inflammatory immune response, which can have a negative effect on living organisms.

The immunomodulatory effect of postbiotics affects not only the cells of the immune system and the digestive tract, but also the central nervous system. Some authors also postulate the possibility of the direct immunomodulatory action of postbiotics [88].

Short-chain fatty acids, which are able to reach the central nervous system by crossing the blood–brain barrier can directly have an impact on the centers in the brain responsible for immune response regulation. In addition to the immunomodulatory effect, postbiotics have a number of other beneficial effects, such as strengthening the barrier of the intestinal mucosa, cytotoxic, and antiproliferative effects [106]. Mack et al. (1999) [107] have shown that postbiotics of the *Lactobacillus plantarum* strain, such as live bacteria, have the ability to reduce the adhesion of pathogenic *Escherichia coli* to intestinal cells. Non-living probiotic microorganisms and their components can compete with pathogens in adhesion to the intestinal epithelial cells, limiting disease processes. Among the active postbiotic components with immunomodulation properties that prevent adhesion and invasion of pathogens and exert potential health-promoting effects, the cell wall fragment of S-layer proteins and bacteriocins should be mentioned. Components of probiotic microorganisms in interaction with intestinal epithelial cells maintain intestinal homeostasis, which has a beneficial effect on the host’s immune system. Postbiotics, in the same way as probiotics, exert potentially beneficial immunomodulatory effects in both acute and chronic inflammatory processes [108].

### 3.4. Safety of Postbiotics

The differences in the mechanism of action and the potential pro-health benefits between probiotics and postbiotics are difficult to assess. In the case of using probiotics among live microorganisms, there are also non-viable postbiotics, as well as their structural components, which should all be classified as postbiotics. In addition, during the passage through the gastrointestinal tract, probiotic microorganisms also die, and digestive processes contribute to the formation of active molecules, which should also be classified as postbiotics. Therefore, the precise comparison is difficult and for clarification there is a need for a head-to-head study of probiotics versus postbiotics. One of the major issues which differentiates probiotics from postbiotics is safety. The non-viable postbiotics are not able to become pathogenic for the host, which is very important, especially for immunocompromised or severely ill patients [109]. The risk of transfer of the antibiotic-resistant properties or pathogenic traits from pathogenic microorganisms to the viable probiotics should be considered [70]. Therefore, according to that, postbiotics seem to be safer to the host than probiotics.

The effect of immunomodulation by selected postbiotics is shown in Table 2.

## 4. Immunobiotics and Post-Immunobiotics as the Components of Functional Food

### 4.1. Immunobiotics in the Aspect of Functional Food

Due to the high market-availability of processed foods, consumers are seeking to make more informed choices about the products they consume, taking into account the safety and composition of consumed products [16]. The conventional products containing components positively affecting health can be recognized as functional food [117]. Generally these products can be classified as: (1) products fortified with ingredients having a positive health influence; (2) raw materials improved/fortified/cleared by changing agricultural practices (i.e., animal feeding and vegetable breeding) or post-harvest treatments (fruits and vegetables); or (3) products cleared from anti-nutritional compounds [118].

The probiotic food is one type of functional food. The health benefits of probiotics, confirmed in this study, have enabled them to be used both in therapy and in prevention, in restoring the natural intestinal microflora, in the production of functional food, and in the preservation of food [119].

Currently, the identification of safe microorganisms with a probiotic potential, that demonstrate a documented and beneficial effect on human health, and are used in the production of functional food and human supplements, has become an important objective in the biotechnology field. The modification of the gut microbiota, enhancement of the epithelial lining barrier-function, competitive adherence to mucosa and epithelium, and modulation of the immune system are some of the health-promoting impacts of favorable gut bacterial cells. The advances in genetic engineering also offer opportunities for the development of new probiotic strains capable of producing targeted metabolites, and showing immunomodulation activity. However, the potential health benefits are attributed to the indicated strain [92,120,121].

### 4.2. Technological Factors in Probiotic Food Development

In order for the health advantages claimed by probiotics to efficiently reach the consumer, the choice of the most suitable strains, culture conditions, and product manufacture are crucial. In the case of using probiotics, there is an additional problem with the development of antibiotic-resistant genes for certain strains, which can consequently lead to the transfer of antibiotic-resistant genes to pathogenic bacteria [122,123].

There are two groups of factors affecting the viability of probiotic bacteria during the production and storage of the final products: food characteristics (acidity, water activity, specific chemical compounds) or storage conditions (moisture, temperature, package permeability to oxygen, time). The consideration of these factors is extremely important during functional food development, because they determine the functionality of the food and boost consumer acceptance of these products. The most popular probiotics found in functional foods and other fermented products include *Leuconostoc*, *Lactobacillus, Enterococcus, Bifidobacterium*, *Streptococcus, Bacillus,* and *Saccharomyces* [92,124,125,126,127].

Some authors point to the fact that the encapsulation process can protect probiotics from outside stresses and result in better bioavailability in the human gastrointestinal tract. The current concept is the co-encapsulation of more than one bioactive compound. This solution may have a synergistic effect, resulting in greater bioactivity and functionality than a single component. The targeted release of active substances through encapsulation is also acknowledged as a good method, and it may be useful in the fight against viral diseases such as COVID-19 [128,129,130].

### 4.3. New Sources of Probiotic Microorganisms

It should be highlighted that the nutrients found in whole diets such as fruits and vegetables act as immune system modulators. Natural compounds of plant origin have a variety of immuno-stimulating capabilities.

Plant-based fermented foods have become an important trend in the food industry, contributing to the global fermented-foods’ trend. These products meet the demand of modern consumers such as those who are lactose-intolerant, vegetarians, or vegans, who are searching for alternatives to supplement their diet as they are unable to eat animal-derived foods [131,132].

A novel trend has recently been seen in vegan probiotic products. However, since the majority of currently used bacterial strains are not obtained from plant-derived matrices, the origin of the strain may compromise the vegan food status. A modern trend has recently been observed in the development of vegan probiotic products [133].

In addition, some authors argue that extending the term “probiotics” to include bacteria isolated from traditionally spontaneously fermented foods seems justified [134,135,136,137]. The microbiota of the environment in which the products were manufactured is composed mainly of microbiota separated from fermented products. They could make an intriguing alternative to gut bacteria if studied, especially in terms of their probiotic capabilities and safety [138].

The data on candidate probiotic bacterial strains of plant origin also show immunomodulatory activities, and are available in the literature [37]. However, the immunomodulatory potential of both animal and plant origin probiotics has been observed, with no clear trend or advantage for others, indicating that the source of origin cannot reflect the ability to induce an immune response.

### 4.4. Functional Food Products with Post-Immunobiotics

The term “postbiotics” also refers to structural probiotic cell fragments, which may have the same beneficial effects on the host as live bacteria. They are a new type of compound able to have an impact on the microbiota. The advantages of using postbiotics are ease of dosing, as well as stability during storage, and their favorable safety profile. They can be used in a controlled and standardized way, while the use of living bacteria depends on the number and metabolic activity of the strain [139].

Postbiotics, as reported by Chaluvadi et al., 2016 [140], could be beneficial as microbial-free food supplements, fermented functional foods, and prophylactic drugs as complementary treatments for a variety of diseases.

The use of postbiotics in food technology may also be seen from the perspective of the packaging process of finished products. An active packaging strategy, which incorporates interactions between the product, packaging material, and environment, is used to extend the food’s shelf life. When delivered to the customer, the active postbiotic packaging technology preserves the product, protecting it from the pathogenic microflora. The postbiotics can be used in a variety of ways to promote stability, including thin coating, an addition to the packaging matrix, and lamination on the polymer [141].

Morniroli et al., 2021 [142], reported that postbiotics can also demonstrate great potential supplements for human health with advantageous effects in pediatric and neonatal disorders. However, further randomized studies are needed to determine which bacterial strain effectively produces beneficial postbiotics, to define the safety profile and recommended doses.

The great advantage of postbiotics is their availability in pure form, their ease of production and storage, and the ability to implement their production on an industrial scale. However, the implementation of the results obtained, in terms of the use of post-immunobiotics in the production of industrial-scale functional food, remains a huge challenge.

#### Aspect of Sensory Quality

An important issue relating to the use of postbiotics in functional food technology is their application for food safety purposes and their effect on the sensory properties of the final product. In order to prevent fungal spoiling, postbiotics from LAB were added to sour semi-hard cheese and cream in Garnier et al.’s (2019) [143] taste analysis. According to the study, postbiotics decreased the fungal population of cheese without having an impact on sensory acceptance, whereas postbiotics at concentrations of 2 and 5% (w/w) had a detrimental impact on the sensory qualities of sour cream.

Consumer acceptance of functional yoghurt supplemented with cape gooseberry (*Physalis peruviana* L.) and containing postbiotics produced by *E. coli* was significantly higher, according to sensory evaluation, for the majority of sensory aspects such as appearance, smoothness, sourness, mouthfeel, and overall acceptance. The product’s color was the sole feature that contrasted poorly with the control sample [144].

On the other hand, Ramos et al. (2022) [145] presented the first study of *Leuconostoc* strains in sheep’s milk producing postbiotic compounds such as aminobutyric acid (GABA), amino acids such as ornithine and tryptophan, and lactic acid. As the authors reported, some chosen bacterial strains might be considered prospective candidates for inclusion in functional dairy products.

## 5. Future Trends and Conclusions

It can be concluded that both pro- and postbiotics have an immunomodulatory effect. Probiotics show health benefits through the modulation of the intestinal microbiome, however some technological limitations reduce their potential for use in the food industry.

In the case of postbiotics, we can talk about intense development on two fronts. In particular, postbiotics, both in the form of dead/inactive cells of probiotics as well as metabolites and components of probiotics, constitute an important development direction in so called functional biotics. The introduction of the definition of this term by ISAP has clarified the previously existing terms. At the same time, we can talk in the future about the development of the functional product segment with its addition.

On the basis of an analysis of the literature, it can be concluded that further testing is required to allow for the use of immunobiotics and post-immunobiotics in the production of functional food. While there are many publications related to the characteristics of probiotic products, there are still few reports available in the literature on specific products supported by post-biotics. Post-immunobiotics appear to be a particularly new development route for functional food.

The postbiotic properties presented indicate that they can contribute to improving human health by providing specific physiological effects, for example through its impact on the immune system, although the precise mechanisms of its operation have not yet been fully explained. New research can help to identify the direction of impact and optimize their production, taking into account stability and storage. It is also necessary to continue interventional clinical trials and to evaluate the whole metabolome.

Postbiotics show huge potential as functional food components, but they also need to be documented in terms of their beneficial effects on human health compared to a specific food matrix. The findings of new biotic response studies of metabolites or host–post-biotic interactions may point to new applications for them in the future.

The production of functional food with ‘biotic’ agents of acceptable quality, adequate nutritional value, and pro-health qualities is becoming a difficult challenge for food producers. Functional food affecting the immune response is the trend of the future. Further understanding of the immunomodulation by food mechanisms is crucial and will allow for the future improvement and development of a new generation of health products.

## Figures and Tables

**Figure 1 microorganisms-11-00104-f001:**
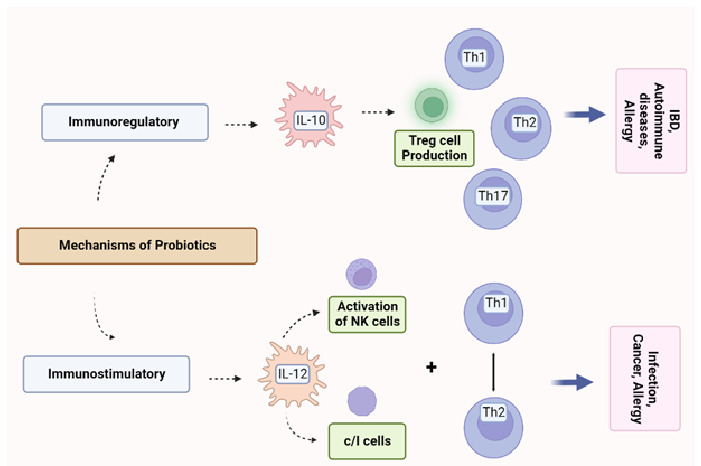
Immune regulation by probiotics.

**Figure 2 microorganisms-11-00104-f002:**
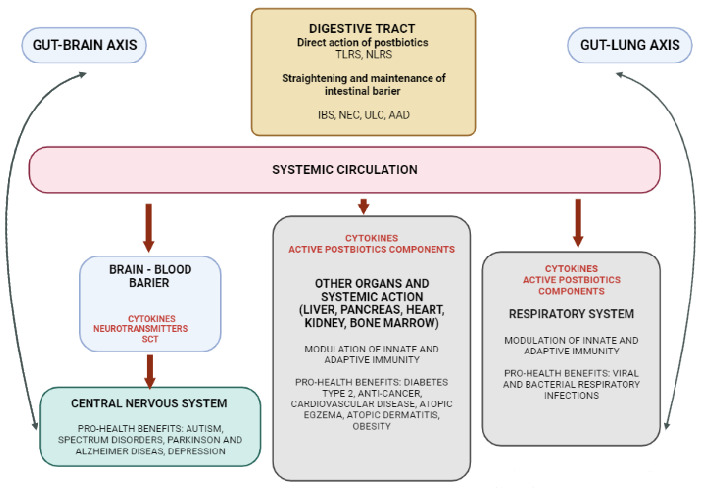
Postbiotics’ mechanism of action. Abbreviations: SCF—Short chain fatty acids; TLR—toll-like receptors; NLRS—nucleotide oligomerization domain-like receptors; ULC—ulcerative colitis; IBS—irritable bowel syndrome; NEC—necrotizing enterocolitis; AAD—antibiotic-related diarrhea.

**Table 1 microorganisms-11-00104-t001:** Effect of immunomodulation by selected probiotics.

Category	Food Components	Research Effect	Reference
**Immunobiotics**			
*Lactobacillus*	*L. delbrueckii* ssp. *bulgaricus* OLL1073R-1 (1073R-1-yogurt)	Consumption of the product affected influenza A virus subtype H3N2-bound Immunoglobulin A (IgA) levels in saliva.	Yamato et al., 2019 [44]
	*L. paracasei* N1115	The intake of yogurt containing *L. paracasei* could protect against the risk of acute upper respiratory tract infection in the middle-aged and elderly; it might be that L. *paracasei* stimulated T-cell immunity	Pu et al., 2017 [45]
	*L. plantarum* NRIC1832 *L. plantarum* NRIC0380	Inhibition of allergy, induction of regulatory T cells by enhancement of IL-10 production, RALDH activity	Noguchi et al., 2012 [46] Yoshida et al., 2013 [47]
	*L. rhamnosus CRL1505* *L. plantarum CRL1506*	Antiviral factors and cytokines/chemokines were increased in lactobacilli-treated PIE cells. The expression of the IL-15 and RAE1 genes that mediate poly (I:C) inflammatory damage was also reduced	Albarracin et al., 2017 [48]
	*L. gasseri OLL2809*	Induction of regulatory T cells	Aoki-Yoshida et. al., 2016 [49]
*Bifidobacterium*	*B. longum MCC1, B. infantis* MCC12, *B. breve* MCC16, *B. pseudolongum* MCC92, *L. paracasei* MCC1375, *L. gasseri* MCC587, *and L. sub* ssp. *lactis* MCC866	*B. infantis* MCC12 and *B. breve* MCC1274 increased the production of INF-β in PIE cells, in response to VR infection. They also increased the expression of CXCL10 and IL-6 genes, especially the *B. infantis*	Ishizuka et al., 2016 [50]
*Saccharomyces*	*S. cerevisiae var. boulardii*	Formation of glutathione, which is responsible for the stimulation of the activity of immune cells.	Badr et al., 2021 [51]

**Table 2 microorganisms-11-00104-t002:** Effect of immunomodulation by selected postbiotics.

Category Probiotic Microorganism	Postbiotic Components	Research Effect	Reference
*L. paracasei* B21060	Cell-free supernatants	Anti-inflammatory effect	Tsilingiri et al., 2012 [110]
*L. paracasei* spp. *paracasei* 06TCa22 and *L. plantarum* 06CC2	Heat-killed cells	Immunomodulation effect	Biswas et al., 2013 [111]
*L. rhamnosus* GR-1	Cell-free supernatants	Immunomodulatory activity	Kościk et al., 2018 [112]
*L. rhamnosus* CRL1505	Peptidoglycan	Improved of Th2 response	Kolling et al., 2018 [113]
VSL#3 (*L. plantarum,* *L. bulgaricus, L. casei* and *L. acidophilus; S. salivarius* *subsp. thermophilus*)	Heat-killed cells	Anti-inflammatory	Sang et al., 2014 [114]
*L. brevis* SBC8803, *L. brevis* 8013 *B. longum*, and *Streptococcus faecalis*	Heat-killed cells	Anti-inflammatory and Enhancement of epithelial barrier permeability	Ueno et al., 2011 [115]
*L. casei* B1	Biosurfactants	Anti-proliferative, anti-oxidative, and anti-adhesion activity against *S. aureus*	Merghini et al., 2017 [116]

## Data Availability

Data are contained within the article.

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
