# Peer review of "Probiotics and Postbiotics as the Functional Food Components Affecting the Immune Response"

_microorganisms, 2022, doi:10.3390/microorganisms11010104_

Round 1
Reviewer 1 Report
Comments(microorganisms-2100284)
This manuscript titled“Probiotics and postbiotics as the functional food components affecting the immune response”provided an interesting overview of the potentials of using probiotics and postbiotics as the functional food components responsible for immune response. However, the well organization of this manuscript is quite required for clear understanding. Additionally, some errors and unclear places existing in the present version should be corrected. Thus, this manuscript has to be modified before resubmitted to the journal Microorganisms.
Specific points are issued as follows.
1/Line 10, what means “your nutritional”? It means human being or consumers?
2/Line 27, please check the error of this sentence.
3/Line 28, please clarify the concrete meaning “high growth”.
4/Line 32-39, references should be presented here.
5/what is logic connection between line 42-43 and line 44-49?
6/Line 59-60, references are necessary.
7/Line 63, here “Microbiota” specifics “intestinal microbiota or others”?
8/Line 71, please check the errors “immunity, [12, 13]..”
9/Line 71-72, “They are carried out, among others, in infectious mucosal and autoimmune pathologies or inflammatory bowel disease”. What it means, and not clear.
10/Line 74-77, references are necessary.
11/Line 95-97, references are necessary.
12/For the part 2, this part is not well-organized for reading,and the important points or opinions should be listed one by one. Like part 3, this part should be organized conceived better, and several subtitles are recommended for this part.
13/Line 154-158, references are necessary.
14/Line 160-164, the two sentences should be combined together because they addressed the probiotic strain-specific which is involved in the immunomodulation. It is why the selected probiotics should be done for immune response.
15/Line 165 for table 1, the column “Food components” should be grouped into Lactobacillus,Bifidobacterium,and Saccharomycesstrains in order.
16/Line 172-175, references are necessary.
17/Line 231-233, references are necessary.
18/Line 235-237, references are necessary.
19/Line 241-244, references are necessary.
20/Line 249-255, what is logic connection between this paragraph and its formers? These post-biotic components included in this paragraph are repeated many times in the different positions of this manuscript (for instance, see line 294-298), or they should be presented in a table?
21/For the Line 310-312 and line 313-322, this two paragraphs specially indicate the bioactivities of probiotics-forming exopolysaccharides, and thus they should be combined together. No sense for separating them.
22/Line 331-342, this paragraph indicates the roles of postbiotics in improving the bio-barrier, and thus it should be presented in a new paragraph.
23/For line 342-344 and line 345-346, what is connection between them?
24/Line 359-360, what is the sense for this sentence because similar words have been expressed several times, or rewrite it.
25/Line 378-380, various mistakes existed in them, and please revise the errors. For instance, “spp.” is not written in italic, and check the mistake spelling of Lactobacilli, Enterococci, and Bifidobacteria as genus. Also, please check errors in the sentence “Bacillus have also been identified 102, 103, 104, 105, 106]”.
26/Line 400-401, references are necessary.
27/Line 423-432, the paragraphs indicate the sensory acceptance of postbiotics as functional components, and thus they should be re-organized.
28/Generally for the part 4, the kinds of functional food should be supplemented here. Additionally, the important points should be listed and classified.
29/Some figures could be supplemented in the text for reading clearly.
30/The formats of references should be unified, such as No.10, 21, 34, and 62.
Author Response
Response to Reviewer 1:
We would like to thank the Reviewer for careful and thorough reading of this manuscript and for the thoughtful comments and constructive suggestions, which help to improve the quality of this manuscript. Our responses to each comment have been marked in blue color below. All the changes in manuscript were marked in colors.
This manuscript titled “Probiotics and postbiotics as the functional food components affecting the immune response” provided an interesting overview of the potentials of using probiotics and postbiotics as the functional food components responsible for immune response. However, the well organization of this manuscript is quite required for clear understanding. Additionally, some errors and unclear places existing in the present version should be corrected. Thus, this manuscript has to be modified before resubmitted to the journal Microorganisms.
Specific points are issued as follows:
Comment 1: 1/Line 10, what means “your nutritional”? It means human being or consumers?
Response: it refers to consumers. The correction has been done.
Comment 2: 2/Line 27, please check the error of this sentence.
Response: the correction has been done (lines: 30-33).
Comment 3: 3/Line 28, please clarify the concrete meaning “high growth”.
Response: We are sorry for a mistake. This expression has been replaced by another one - “highly developed” (Line 32).
Comment 4: 4/Line 32-39, references should be presented here.
Response: thank you for this suggestion. The suitable references were added (references no.: 2-6, line 36).
Comment 5: 5/what is logic connection between line 42-43 and line 44-49?
Response: thank you very much for this question. It helped us to improve this section of manuscript. These two paragraphs were connected. Firstly, the term „functional food” was characterized. Then a reference was made to various food components which can affect the immune response.
Comment 6: 6/Line 59-60, references are necessary.
Response: the reference was completed (reference no. 16, line 64).
Comment 7: 7/Line 63, here “Microbiota” specifics “intestinal microbiota or others”?
Response: thank you for this suggestion. The correction has been done (line 67).
Comment 8: 8/Line 71, please check the errors “immunity, [12, 13]..”
Response: the correction has been done (line 75).
Comment 9: 9/Line 71-72, “They are carried out, among others, in infectious mucosal and autoimmune pathologies or inflammatory bowel disease”. What it means, and not clear.
Response: thank you for this comment. This part of manuscript was deleted. It was replaced by another sentence (lines: 76-78): “It has been reported that the dysbiosis play some role in the pathogenesis of many diseases such as gastrointestinal, cardiovascular diseases obesity or diabetes [20, 21, 22, 23].”
Comment 10: 10/Line 74-77, references are necessary.
Response: the reference was completed (references no. 23, 24, line 83).
Comment 11: 11/Line 95-97, references are necessary.
Response: the references were completed (references no. 29,30; line 103).
Comment 12: 12/For the part 2, this part is not well-organized for reading,and the important points or opinions should be listed one by one. Like part 3, this part should be organized conceived better, and several subtitles are recommended for this part.
Response: The corrections have been done according to the reviewer’s suggestion (subsection no. 2, 3, 4).
Comment 13: 13/Line 154-158, references are necessary.
Response: the references were completed (references no. 38, 39, line 172).
Comment 14: 14/Line 160-164, the two sentences should be combined together because they addressed the probiotic strain-specific which is involved in the immunomodulation. It is why the selected probiotics should be done for immune response.
Response: thank you for this suggestion. In the revised version of the manuscript, the sentence “The previous works regarding the effect of immunomodulation by probiotics as food components are shown in Table 1” was moved right above Table 1, due to forming a new subsection called “Effect of immunomodulation by selected probiotics”.
Comment 15: 15/Line 165 for table 1, the column “Food components” should be grouped into Lactobacillus,Bifidobacterium,and Saccharomyces strains in order.
Response: thank you for this suggestion. The correction of the Table 1 has been done.
Comment 16: 16/Line 172-175, references are necessary.
Response: the corrections have been done (references: no. 54-58, lines: 224-225; no. 59, line 230).
Comment 17: 17/Line 231-233, references are necessary.
Response: the references (no.84; 85) were completed (Line:292).
Comment 18: 18/Line 235-237, references are necessary.
Response: the reference no. 88 was added (Line: 298).
Comment 19: 19/Line 241-244, references are necessary.
Response: the reference no. 83 was added (Line: 305).
Comment 20: 20/Line 249-255, what is logic connection between this paragraph and its formers? These post-biotic components included in this paragraph are repeated many times in the different positions of this manuscript (for instance, see line 294-298), or they should be presented in a table?
Response: The paragraph was changed according to the reviewer’s suggestion. The Figure 2 was added.
Comment 21: 21/For the Line 310-312 and line 313-322, this two paragraphs specially indicate the bioactivities of probiotics-forming exopolysaccharides, and thus they should be combined together. No sense for separating them.
Response: The paragraph was changed according to the reviewer’s suggestion (lines: 377-386).
Comment 22: 22/Line 331-342, this paragraph indicates the roles of postbiotics in improving the bio-barrier, and thus it should be presented in a new paragraph.
Response: The paragraph was changed according to the reviewer’s suggestion (Lines: 394-396).
Comment 23: 23/For line 342-344 and line 345-346, what is connection between them?
Response: The part of the manuscript was completed according to the reviewer’s suggestion. Lines 345 -346 were deleted (Line 409).
Comment 24: 24/Line 359-360, what is the sense for this sentence because similar words have been expressed several times, or rewrite it.
Response: thank you for this opinion. This sentence was deleted.
Comment 25: 25/Line 378-380, various mistakes existed in them, and please revise the errors. For instance, “spp.” is not written in italic, and check the mistake spelling of Lactobacilli, Enterococci, and Bifidobacteria as genus. Also, please check errors in the sentence “Bacillus have also been identified 102, 103, 104, 105, 106]”.
Response: thank you for this note. The corrections have been done (Line: 475).
Comment 26:26/Line 400-401, references are necessary.
Response: the references (no.138-141, lines: 498-499) were completed.
Comment 27:27/Line 423-432, the paragraphs indicate the sensory acceptance of postbiotics as functional components, and thus they should be re-organized.
Response: thank you for this suggestion. The corrections have been made. The section no. 4 was divided into 4 subsections:
4.1 Immunobiotics in the aspect of functional food (Line: 438)
4.2 Technological factors in probiotic food development (Line: 462)
4.3 New sources of probiotic microorganisms (Line: 484)
4.4 Functional food products with postimmunobiotics (Line: 509)
4.4.1 Aspect of sensory quality (Line: 537).
Comment 28:28/Generally for the part 4, the kinds of functional food should be supplemented here. Additionally, the important points should be listed and classified.
Response: The paragraph no. 4 was changed according to the reviewer’s suggestion (references no : 119-120; lines: 446-451).
Comment 29: 29/Some figures could be supplemented in the text for reading clearly.
Response: thank you for this suggestion. The Figure 1 and Figure 2 were added to the revised text of the manuscript.
Reviewer 2 Report
This paper summarizes the effects of probiotics and postbiotics on immune response. The article is clearly written and makes sense in conjunction with the latest research. But some problems need further improvement.
1. When the strain name appears for the first time, it should be written in full Latin, and when it appears for the second time, it should be written in short Latin.
2. This manuscript should list some probiotics and explain their sources and effects.
3. This manuscript should further produce the reaction mechanism of immune response, and it is necessary to show the immune regulation process of probiotics and postbiotics in the form of pictures.
4. Whether the mechanism of action of different probiotics and postbiotics is different should be reviewed.
5. The reference format of this manuscript needs to be unified.
Author Response
Response to Reviewer 2:
We would like to thank the Reviewer for careful and thorough reading of this manuscript and for the thoughtful comments and constructive suggestions, which help to improve the quality of this manuscript. Our responses to each comment have been marked in blue color below. All the changes in manuscript were marked in colors.
Comment 1:
This paper summarizes the effects of probiotics and postbiotics on immune response. The article is clearly written and makes sense in conjunction with the latest research. But some problems need further improvement.
When the strain name appears for the first time, it should be written in full Latin, and when it appears for the second time, it should be written in short Latin.???
Response: it refers to consumers.
Comment 2: This manuscript should list some probiotics and explain their sources and effects.
Response: We thank you for this comment. This information we included in Table 1.
Comment 3: This manuscript should further produce the reaction mechanism of immune response, and it is necessary to show the immune regulation process of probiotics and postbiotics in the form of pictures.
Response: we are grateful for this suggestion. The suitable figures (no. 1; 2) were added in the text of manuscript.
Comment 4: Whether the mechanism of action of different probiotics and postbiotics is different should be reviewed.
Response: these mechanisms were reviewed for probiotics: Lines:181-208, Figure 1) and for postbiotics: Lines and also shown in Figure 2 (Lines: 328-350). In addition, the difference in mechanisms of immunomodulatory actions between probiotics and postbiotics are shown in lines: 358-360.
Comment 5:The reference format of this manuscript needs to be unified.
Response: the corrections have been done.
Reviewer 3 Report
My issue with this paper is the lack of depth and focus. The authors have tackled a potentially large area, probably too big. However, the paper is really mainly an overview and not a review. A lot of small sections which mainly focus on providing definitions and/or genetic statements on terminology and definitions, but scientific depth is lacking throughout the paper. The paper reads like an introductory chapter to a thesis or something. In that case, it serves the purpose well. However, it is, in my opinion, not a scientific review paper. Merely a broad overview.
Author Response
Response to Reviewer 3:
My issue with this paper is the lack of depth and focus. The authors have tackled a potentially large area, probably too big. However, the paper is really mainly an overview and not a review. A lot of small sections which mainly focus on providing definitions and/or genetic statements on terminology and definitions, but scientific depth is lacking throughout the paper. The paper reads like an introductory chapter to a thesis or something. In that case, it serves the purpose well. However, it is, in my opinion, not a scientific review paper. Merely a broad overview.
Response: We would like to thank the Reviewer for the careful and thorough reading of this manuscript and for the constructive suggestions, which help to improve the quality of this manuscript.
Round 2
Reviewer 1 Report
No more comments.
Reviewer 2 Report
The manuscript has been revised.